Using PLUS-InVEST-OPGD model to explore spatiotemporal variation of ecosystem carbon storage and its drivers in Jinsha river basin, China

Huang Lichang 1 2 3
Ding Xue 4228@ynnu.edu.cn dingxue_1983@163.com 1 2 3
Wang Jinliang 1 2 3
Peng Shuangyun 1
1 Faculty of Geography, Yunnan Normal University , Yunnan , Kunming , China
2 Key Laboratory of Resources and Environmental Remote Sensing for Universities in Yunnan Kunming, Yunnan Normal University , Kunming , Yunnan Province , China
3 Center for Geospatial Information Engineering and Technology of Yunnan Province, Yunnan Normal University , Kunming , Yunnan Province , China
Brygadyrenko Viktor
Electronic publication date: 2025 Jul 28
Publication date: 2025
Volume: 13
Electronic Location ID: e19681
Received 2025 Feb 11; Accepted 2025 Jun 10
Copyright: ©2025 Huang et al.
Copyright year: 2025
Copyright holder: Huang et al.
License: This is an open access article distributed under the terms of the Creative Commons Attribution License, which permits unrestricted use, distribution, reproduction and adaptation in any medium and for any purpose provided that it is properly attributed. For attribution, the original author(s), title, publication source (PeerJ) and either DOI or URL of the article must be cited.
License URL: https://creativecommons.org/licenses/by/4.0/

Keywords: Land Use and Cover Change (LUCC), Carbon storage, Multi-scenario simulation, Future scenario simulation, Driving mechanism, Jinsha River Basin in Yunnan Province, PLUS-InVEST model

Funding: The Key Laboratory of Remote Sensing in Higher Education Institutions in Yunnan Province Yunnan Provincial Engineering Research Center for Geographic Spatial Information Science and Technology Major Project of Yunnan Province 202302AO370003 Basic Research Project of Yunnan Province 202301AT070173 Yunnan Fundamental Research Projects 202401AT070103 This work was funded by The Key Laboratory of Remote Sensing in Higher Education Institutions in Yunnan Province and the Yunnan Provincial Engineering Research Center for Geographic Spatial Information. This research was also funded by Science and Technology Major Project of Yunnan Province (Science and Technology Special Project of Southwest United Graduate School - Major Projects of Basic Research and Applied Basic Research): Vegetation change monitoring and ecological restoration models in Jinsha River Basin mining area in Yunnan based on multi-modal remote sensing (Grant No.: 202302AO370003), Basic Research Project of Yunnan Province, Project name: Identification of high altitude Remote geological hazards in mountainous areas of Western Yunnan based on Multi-source Remote Sensing Technology - A case study of the Ratuti River Basin (NO. 202301AT070173), Remote sensing estimation of aboveground carbon sink of vegetation in the central Yunnan urban agglomeration and its response to climate change and human activities, supported by Yunnan Fundamental Research Projects (grant NO. 202401AT070103). The funders had no role in study design, data collection and analysis, decision to publish, or preparation of the manuscript.

==============================
Land-Use/Land-Cover Change (LUCC) is a key disturbance factor of the carbon cycle in terrestrial ecosystems, and the study on the coupling mechanism between LUCC and carbon storage is of great scientific value for implementing a regional carbon-neutral strategy. In this study, the Jinsha River Basin in Yunnan Province, which has outstanding ecological vulnerability, is taken as the research object, and a synergistic analytical framework of “spatial and temporal pattern drivers” is constructed by integrating multi-temporal remote sensing data and multi-model coupling method. Based on the high-precision 30 m land use data from 1990 to 2020, the PLUS-InVEST-OPGD multi-model coupled system was used to simulate and predict the characteristics of spatial and temporal carbon storage differentiation in 2030 under four development scenarios, namely, natural development (ND), ecological protection (EP), farmland protection (FP), and economic development (ED), and to analyze the driving mechanism using the Optimal Parameter Geodetic Probe (OPGP). The driving mechanism is analyzed using an optimal parameter geodetector. The main findings were: (1) The land use structure of the watershed in the study area showed a significant ecological-productive dichotomy, with forest land (60.58%), grassland (28.85%) and cultivated land (7.19%) constituting the core carbon sink carriers (the average proportion of which was 96.62% from 1990 to 2020). Still, the area of forest and grassland decreased by a total of 2,757.84 km2 in the past 30 years, and the expansion of construction land amounted to 2,321.91 km2; (2) the spatial and temporal evolution of carbon storage shows the heterogeneous characteristics of “overall decreasing and local optimization”, in which the carbon loss from forest to grassland conversion is as high as 30% of the total carbon loss, and the expansion of construction land leads to irreversible decay of carbon sinks of about 50%; (3) a multi-scenario simulation shows that the EP scenario minimizes the loss of carbon storage (−2.46 × 106 t) by maintaining a 96.82% ecological land share in 2030, reducing the carbon deficit by 7.79 × 106 t compared with the ND scenario; (4) the average annual temperature is the largest single factor affecting carbon storage, and its interaction with the population factor has a high q value of 0.84. This study innovatively reveals the nonlinear threshold effect of LUCC-carbon storage response in the Jinsha River Basin of Yunnan Province, and the proposed optimization model of “ecological protection” can provide decision support and corresponding reference for the construction of ecological security barriers in the upper reaches of the Yangtze River.

Introduction

Among the many research topics on the global carbon cycle, the issue of carbon storage changes in terrestrial ecosystems has occupied a central position for decades and has received close attention from academia and practice (Newbold et al., 2015). The Sixth Assessment Report of the Intergovernmental Panel on Climate Change (IPCC) particularly pointed out that 2007 to 2016, global Land use change (LUCC) contributed about 23% of anthropogenic carbon emissions from 2007–2016, making it the second largest carbon source after fossil fuel combustion (Lambin et al., 2001). Relevant research results show that land use change directly affects the change of carbon storage in terrestrial ecosystems (Liu et al., 2023). When ecological lands such as forests and grasslands are converted to construction land, the large-scale removal of vegetation and the turning over of soils lead to the large-scale release of carbon stored in trees and soils to the atmosphere in the form of carbon dioxide, which leads to the reduction of carbon storage in ecosystems (Liu et al., 2019). On the contrary, if the construction land can be restored to ecological lands such as forest land and grassland, the carbon storage capacity can be significantly improved and contribute to mitigating global climate change (Chazdon & Brancalion, 2019). With the continuous acceleration of urbanization, the world faces the serious challenge of constantly reducing carbon storage (Weiskopf et al., 2024). Therefore, an in-depth investigation of the impacts of land use change on the spatial and temporal changes of carbon storage in terrestrial ecosystems and their driving mechanisms is vitally practical for the development of scientific and reasonable ecosystem protection and restoration strategies, optimization of regional ecosystem service functions, and the formulation of policies for sustainable socio-economic development (Li et al., 2013; Mallapaty, 2020; Zhao et al., 2022).

As integral units of “natural-social” complex ecosystems, watershed carbon storage dynamics have become a frontier of global change research. Evidence suggests that areas of river corridors (active channels and floodplains) hold more organic carbon than the surrounding terrestrial ecosystems (Wohl & Knox, 2022). Existing studies have made breakthroughs in four dimensions: (1) At the level of research subjects, from early single-landclass carbon density measurements, Seifu et al. (2021) analyzed the elevation gradient of surface soil organic carbon in the northern watershed of Ethiopia; and Gai, Xu & Du (2023) focused on cultivated land-use transition focusing on the carbon storage in the Songhua River basin of the Chinese Black Soil Zone differentiation characteristics, etc.) developed into a coupled simulation system with multiple objects and models. (2) In the value dimension, Rajbanshi & Das, (2021) pioneered the incorporation of economic valuation into the study of carbon storage in the Konar Basin and constructed a two-dimensional evaluation framework of “ecology-economy”. (3) In the temporal dimension, Li et al. (2023c) demonstrated through the case study of the Iloilo River Basin, that the coupling of historical inversion and scenario prediction can improve the foresight of carbon management strategies. (4) In terms of driving force analysis, the “carbon pool response coefficient method” proposed by Davoudabadi et al. (2024) realizes the quantitative decoupling of natural and anthropogenic drivers. It provides a new paradigm for the assessment of watershed carbon sink function. Although the above studies have made significant progress, there are still three bottlenecks in the current research: (1) Low prediction accuracy: Mainstream models such as CLUE-S (Feng et al., 2023) have low prediction accuracy due to the limitation of linear transformation rules. CA-Markov (Yulianto, Maulana & Khomarudin, 2019) has a significantly decreased simulation accuracy in complex terrain compared to plains. FLUS (Xu et al., 2022) introduces the BP neural network, but it is still difficult to deal with the nonlinear response in the alpine and canyon areas; (2) Insufficient analysis of driving mechanisms: existing studies mostly focus on the natural description of carbon storage changes (e.g., Li et al.’s (2023b) statistical analysis of the Three Gorges Reservoir area) but ignore the deeper driving mechanisms. There is a lack of systematic study of the hidden driving force of social factors such as GDP; (3) Lack of analysis of long time-series studies: current studies have significant time-series limitations, e.g., Liao et al. (2024) analyzed the carbon storage situation in the Pearl River Delta (PRD) based on land use change from 2005 to 2020, which is difficult to capture the lagged response of carbon storage to LUCC.

Based on this, this study focuses on the Jinsha River Basin in Yunnan Province, which is characterized by outstanding ecological vulnerability and acute human-land conflicts and where drastic land use/cover changes due to frequent human activities have profoundly impacted carbon storage changes. Wang et al. (2025) showed that the overall spatial distribution of carbon storage in the Jinsha River dry heat river valley from 1995 to 2020 was characterized by “high in the center, low in the surroundings, and higher in the west than in the east”, and the decrease in the area of forest and grassland was the main reason for the decrease in carbon storage. In the same vein, Wen et al. (2024) showed that in the Jinsha River Basin of Yunnan Province, the loss of carbon storage in the eastern part of the spatial distribution in 2000, 2010, and 2020 was more severe than that in the western part of the basin, mainly because of the high intensity of human activities and low vegetation cover. This spatial heterogeneity is mainly attributed to the gradient difference between human activity intensity and the spatial differentiation of vegetation cover. To explore this issue in depth, we need to further simulate and predict the corresponding carbon stock in the Jinsha River Basin (Yunnan section) under different conditions, and this study innovatively constructs a multi-model coupled modeling framework of “PLUS-InVEST-OPGD”. First, the PLUS model effectively avoids the traditional modeling dilemma of exponential growth in the number of transformation types when the number of categories increases by integrating the transformation and pattern analysis strategies (Zhang, Liao & Sun, 2024). It is verified that the model simulation accuracy reaches more than 80% (Kappa = 0.85), significantly better than the traditional CA-Markov and FLUS models (Wang et al., 2022a). Second, the InVEST (Integrated Valuation of Ecosystem Services and Trade-offs) model (Sharp et al., 2018) developed by Stanford University was used to estimate carbon storage. The model has the following advantages: (1) it supports multi-scale carbon storage assessment and can be applied from global to watershed scale; (2) it has a scenario simulation function and can quantify carbon storage changes under different land use patterns; (3) it can effectively analyze the impact mechanism of human activities on carbon storage. Finally, the optimized parameter geodetector (OPGD) algorithm was introduced for driver analysis (Zhang et al., 2022). Compared with the traditional geodetector, this algorithm significantly improves the resolution accuracy of the association relationship between driving factors and carbon storage by (1) optimizing the method of determining the number of classifications, (2) improving the spatial partitioning strategy, and (3) refining the interaction detection mechanism (Song et al., 2020), which effectively solves the scientific problem of insufficient resolution of the mechanism in the previous studies.

Based on the above methodological system, this study constructed a multi-scenario optimization simulation framework and realized the high-precision simulation of carbon storage’s spatial and temporal dynamics in the watershed. The results of this study provide a new analytical paradigm for understanding the coupling mechanism of LUCC and carbon storage and a scientific basis for decision-making on ecological protection and sustainable development of watersheds.

Study Area and Data Sets

Overview of the study area

The Jinsha River Basin in Yunnan Province is located in the southwestern part of China and the northern location of Yunnan Province. It is approximately 1,560 km long in Yunnan Province (Fig. 1). The entire study basin is located between 98°20′– 105°19′E longitude and 24°13′– 29°16′N latitude, occupying nearly one-third of the total land area of Yunnan Province. The basin’s complex topography, variety of climate types, elevation differences, and uneven distribution of precipitation conditions combine to create its unique climatic characteristics. In the past 30 years, the basin’s social development has been rapid, land use has undergone drastic changes, and ecosystem carbon storage has particularly fluctuated. A comprehensive understanding of the interplay among these intricate factors within the carbon cycle is crucial for developing effective environmental protection strategies and fostering the sustainable development of the basin’s ecological environment.

Figure 1 Location map of Jinsha River Basin in Yunnan Province.

(A) Yunnan Province on the Chinese border; (B) Jinsha River Basin in Yunnan Province and abbreviations of various states and cities; (C) specific scope of Jinsha River Basin in Yunnan Province.

Data set

Land use data

In this paper, the four-period land use data of the Jinsha River Basin in Yunnan Province for the intervals of 10 years in 1990, 2000, 2010, and 2020 were obtained from the first set of global 30-meter resolution land cover dynamic products (GLC_FCS30D) released by the team of Liu Liangyun, a researcher at the Institute of Aerospace Information Research Institute, Chinese Academy of Sciences from 1985 to 2022, with a spatial resolution of 30 m (Zhang et al., 2024). For subsequent in-depth analyses, the data were reclassified into the six major categories in the first-level class, including cultivated land, forest, grassland, water body, construction land, and unused land.

Natural and socio-economic drivers

Since land-use change has become a key driver affecting carbon storage in terrestrial ecosystems, land-use change is mainly influenced by the interaction between human activities and ecosystems (Canadell & Raupach, 2008). Therefore, selecting appropriate drivers to model and predict future land use change and its impact on carbon storage is significant. However, there are substantial limitations in the screening and quantification of drivers in existing studies, and the selection of drivers is one-sided: traditional models mostly focus on natural or socio-economic one-dimensional factors, ignoring the synergistic effects of the coupled “nature-society” system (Pyles et al., 2022; Walden et al., 2023). In this study, we address the above problem by constructing a multidimensional driving factor system, integrating 12 categories of high contribution factors, and the contribution value share of the driving factors is positively correlated with their contribution, i.e., the higher the share, the stronger their driving force on landform evolution (Du et al., 2023). As shown in Fig. 2 and Table 1, there are four dimensions involved: topographic factor, climatic factor, socio-economic factor, and distance factor. Topographic factors include slope (X6), DEM (X11), and soil type (X12); climatic factors are annual precipitation (X2) and annual temperature (X7); population (X3) and GDP (X9) for socio-economic factors; and distance factors include distance from highway (X1), railroad (X4), water system (X5), residential area (X8), and the secondary roads (X10).

Figure 2 Twelve types of driving factors visualization.

Research framework

In this paper, the land use data of the study area for the years 1990–2020 were collected and pre-processed to synthesize the area values of various types of land expansion. Using the existing detailed land type dataset, the Markov chain module (Zou et al., 2023) was introduced to estimate the land development demand under different scenarios. The future planning and development policies of the Jinsha River Basin (Yunnan section) were considered. The restricted conversion areas, such as nature and water resource reserves, were established, and the restricted conversion rules for special land types were constructed (Fu et al., 2024) to simulate the land use patterns under different scenarios in the future 2030. The overall technical framework of the study is shown in Fig. 3.

Research Methodology

InVEST model

The carbon module of the InVEST model calculates changes in ecosystem carbon storage due to land use change, which is important for understanding the global carbon cycle and addressing climate change (Al Kafy et al., 2023). The model has been widely used due to its ability to geographically integrate the evolution of land use change and terrestrial ecosystem carbon storage dynamics and effectively elucidate the relationship between these changes (Huang et al., 2023). This experiment followed the classification rules of ecosystem carbon storage. It used four basic carbon pools: above-ground biocarbon density, below-ground biocarbon density, soil biocarbon density, and dead biocarbon density (Wang et al., 2022b). The carbon density values of each category and type were first taken from the nationwide carbon pools of previous related studies (Table 2). Then, the average annual precipitation and average annual temperature data of the study area and the whole country were utilized to correct the carbon density (Guo et al., 2023; Liu et al., 2024) to obtain the specific carbon density values of the Jinsha River Basin area in Yunnan Province. The carbon density of dead organic matter was referred to in the research data of Wusimanjiang et al. (2024) without correction. Finally, the carbon density values of different land use types in the Jinsha River Basin were calculated (Table 2).

Table 1 Basic data for the study basin.

Type	Name	Resolution/ Format	Source	Purpose	
Land use type	Land use data at 30 m resolution for the Jinsha River basin, 1990, 2000, 2010, and 2020	30 m/tif	International Research Center of Big Data for Sustainable Development Goals-CBAS (https://data.casearth.cn/thematic/glc_fcs30/314)	Extract the area of the land class and the necessary data for the analysis of the PLUS and InVEST models	
Natural conditions	Temperature	1,000 m/tif	The China Meteorological Data Service Center (https://data.cma.cn/en)	Driver data (Participate in the PLUS model ground class expansion analysis and one of the independent variable factors of the OPGD model), and used to correct the calculation of carbon density in the study area	
Precipitation	1,000 m/tif	The China Meteorological Data Service Center	
DEM	30 m/tif	Geospatial Data Cloud (https://www.gscloud.cn/)	Driver data (Participate in the PLUS model ground class expansion analysis and one of the independent variable factors of the OPGD model)	
Slope	30 m/tif	Slope analysis on DEM	
Soil type	1,000 m/tif	Harmonized World Soil Database version (HWSD) (https://www.fao.org/soils-portal/soil-survey/soil-maps-and-databases/harmonized-world-soil-database-v12/en/)	
Socio-economic	Population	1,000 m/tif	The China Resources and Environmental Science Data Center	
GDP	1,000 m/tif	
Traffic conditions	Distance from the secondary road	/tif	National Catalogue Service For Geographic Information (https://www.webmap.cn)	
Distance from highway	/tif	
Distance from railway	/tif	
Distance from the water system	/tif	
Distance from the residential area	/tif	

Figure 3 Overall technology road map.

Table 2 Carbon density of different land use types (t/hm2).

Land use type	Cabove	Cbelow	Csoil	Cdead	
Cultivated land	8.97b/47.46	11.12a/58.83	108.40c/124.85	2.11a	
Forest	49.40d/56.89	18.40a/21.19	158.80c/182.89	2.78a	
Grassland	6.56b/7.56	86.5e/99.62	99.9c/115.06	2.42a	
Water body	0.36d/1.90	0f	0f	1.78a	
Construction land	2.93d/15.50	0f	34.36a/39.57	0a	
Unused land	0.30b/1.59	0f	0f	0.96a	
Notes.

a Cited from Wusimanjiang et al. (2024).

b Cited from Chen, Liu & Li (2002).

c Cited from Jia & Hu (2024).

d Cited from Zhao et al. (2024).

e Cited from Mi et al. (2023).

f Cited from Li et al. (2023d).

Primary carbon density/revised carbon density.

PLUS model

The patch-generating land use simulation (PLUS) model (Liang et al., 2021) is a simulation system based on the patch-generation mechanism of land use change, and its core consists of two major strategies: the Land Expansion Analysis Strategy (LEAS) and the CA model based on multi-class stochastic patch seeds (CARS). The model realizes the prediction of land use demand by integrating the random forest algorithm and adaptive sampling technique, as well as mining dynamic neighborhood weights and adaptive transformation rules, which effectively solves the problem of decreasing the simulation accuracy of the traditional CA model in complex terrain areas (Huang et al., 2023). The weights of each category of cultivated land, forest, grassland, water body, construction land, and unused land for the study were finally decided through auxiliary calculations and set as follows: 0.2045, 0.2483, 0.4313, 0.0141, 0.1011, and 0.0007. Regarding accuracy control, the PLUS model can better capture the dynamics of land use change and provide a more realistic and accurate simulation (Liang et al., 2021; Tian et al., 2022). In this study, the distribution of land types in 2020 was simulated by using two historical land use type atlases in 2000 and 2010, which were exhaustively compared and validated with the actual land use data in 2020 (Fig. 4). The validation results show that the overall accuracy reaches 0.92. The Kappa coefficient is 0.85, which meets the accuracy requirements needed for the experiment (Jiang et al., 2024). Based on the above validation results, the prediction of the land use distribution pattern in 2030 can be carried out.

Figure 4 Comparison of land use simulation of study area in 2020.

Given the long-term planning for the future development of the Jinsha River Basin in Yunnan Province, four distinct development scenarios were designed in this study to comprehensively explore the land-use changes and their impacts on carbon storage under different development paths, with the specific probability adjustments of the transfer rules as follows: Natural development (ND) scenario: This scenario was used as the control group without conversion. According to the requirements of the “Special Supervision Action for the Protection of the Ecological Environment and Resources of the Jinsha River Basin (Yunnan Section),” the water resources area in the study area is strictly prevented from shrinking in scope. It is regarded as a restricted development watershed. Ecological protection (EP) scenario: According to the “Three Screens, Two Belts, Six Corridors and Multiple Points” ecological security pattern delineated in the “Yunnan Provincial Land Spatial Planning (2021–2035)”, it is prohibited to add new non-essential construction land in sensitive areas such as ecological barrier zones and the Jinsha River dry and hot river valley belt. The transfer of forest and grassland to construction land will be reduced by 40%, the transfer of cultivated land and water body to construction land will be reduced by 30%, and the transfer of unused land and construction land to forest land will be increased by 20%. In comparison, the construction land transfer to grassland will be increased by 10%. In addition, water resources and nature reserves in the study area need to be used as restricted development areas to emphasize ecological protection. Farmland protection (FP) scenario: The program strictly abides by the “three control lines”, prioritizes the guarantee of basic cultivated land space within the Jinsha River Basin, and prohibits the occupation of high-quality cultivated land for non-agricultural construction by the planning requirements. The flow of cultivated land to construction land is reduced by 70%, the flow to grassland and water body is reduced by 40%, and the flow of unused land to cultivated land is increased by 50%. This plan not only extracts long-term stable cultivated land and high-quality cultivated land that meets slope requirements within the research area but also uses it as a restricted conversion area with water resources. Economic development (ED) scenario: To promote the economy of the basin at a high level, especially the Jinsha River Basin (Yunnan section), and to encourage the construction of the “Diversion of water from Dianzhong” project, it is planned that 100 new major water conservancy construction projects will be initiated, which will allow for the construction of water conservancy industries in the relevant areas and the development of economic construction. The scenario reduces the transfer of construction land to forest land, grassland, water body and unused land by 20%, while the transfer of cultivated land, forest land and grassland to construction land increases by 20%.

The optimal parameter geographic detector

Geodetector is a new model for detecting spatial dissimilarity created by Wang & Xu (2017), which is a new statistical method for detecting the consistency of the spatial distribution pattern of dependent and independent variables based on the theory of geospatial dissimilarity (Liang et al., 2018). The Optimal Parameter Geodetector (OPGD) model developed by Song et al. (2020) is based on traditional detectors by identifying and optimizing the parameters to improve the accuracy and efficiency of detection and analysis of conventional probes and solve the problem of discrete methods with a limited number of bins. Among them, the probe value q is a measure of the strength of the explanatory power in the range of [0,1]; the larger the value, the stronger the explanatory power. The OPGD model supports two major modes: factor detection and interaction detection. The q statistic of the factor probes compares the observed variance of the whole study watershed with that within the variable stratum (Wang et al., 2010), and this comparison helps to assess the degree of variability and the spatial distribution pattern of the data. On the other hand, the interaction detector method is suitable for analyzing the interaction of two random drivers whose q values jointly explain the strength of spatial and temporal variability in carbon storage, i.e., q(X1∩X2). The interaction detector explores five types of interactions, namely, non-linear weakening, univariate weakening, bivariate strengthening, mutual independence, and non-linear strengthening (Wang, Zhang & Fu, 2016).

Results

Changes in land use in the study area, 1990–2020

From 1990 through 2020, the spatial pattern of land use in the study area showed diversified characteristics, with vast forest and grassland covering most of the basin. Cultivated land is scattered, mainly concentrated in central Yunnan’s lake basin plateau area and the middle-low mountain broad valley basin in southwestern Yunnan. The distribution of construction land is closely related to that of cultivated land. In addition, construction land tends to be clustered and distributed around the watershed, especially in urban centers such as Dianchi Lake in Kunming City and the Erhai Sea in Dali (Fig. 5).

Figure 5 Distribution pattern of land use in Jinsha River Basin, Yunnan Province, 1990–2020.

According to the analysis of land use data of the Jinsha River Basin in Yunnan Province from 1990 to 2020 (as shown in Table 3), the Jinsha River Basin in Yunnan Province presents a typical “three main and three auxiliary” land use pattern. In the lowest percentage of area of each category during the 30 years, forest land (60.58%), grassland (28.85%) and cultivated land (7.19%) together constituted the main body of the research area watershed (combined >96.62%), while the secondary categories of water body (0.88%), construction land (0.61%), and unused land (0.08%) accounted for the lowest combined share of only 1.58%, but with significant ecological Sensitivity. Over the past 30 years, land use changes have changed differently. During the period of dramatic transition from 1990 to 2000, cultivated land and grassland faced a crisis of substantial reduction: cultivated land area decreased by 253.34 km2 (K = −0.20%) and grassland by 1,840.00 km2 (K = −0.35%), while land for construction exploded in size, expanding by 1.02 times of the original base (K = −10.21%), showing the characteristics of “enclave-style” spatial expansion. During the smooth adjustment period from 2000 to 2010, forest land was slightly degraded, with an area decrease of 1,459.21 km2 (K = −0.14%), and the growth rate of construction land slowed down. During the new type of contradiction period from 2010 to 2020, the grassland continued to be lost, with a cumulative decrease of 993.77 km2 (K = −0.20%), of which 44.04% was converted to construction land.

Table 3 Land use area and single-movement attitudes in the study area, 1990–2020.

Land use type	Area/km2	Single dynamic degree of land use/% (K value)	
	1990	2000	2010	2020	1990–2000	2000–2010	2010–2020	1990–2020	
Cultivated land	12,693.06	12,439.72	12,584.04	12,833.26	−0.20	0.12	0.20	0.04	
Forest	105,344.17	106,256.79	104,797.58	104,903.43	0.09	−0.14	0.01	−0.01	
Grassland	52,224.61	50,384.61	50,901.28	49,907.51	−0.35	0.10	−0.20	−0.15	
Water body	1,528.49	1,607.13	1,617.62	1,798.51	0.51	0.07	1.12	0.59	
Construction land	1,060.80	2,144.19	2,945.03	3,382.71	10.21	3.73	1.49	7.30	
Unused land	166.63	156.68	143.66	164.70	−0.60	−0.83	1.47	−0.04	

Based on the land use area transfer matrix from 1990 to 2020 (as shown in Table 4 and Fig. 6), analyzing from the transfer out level, the cultivated land system is vulnerable, with a cumulative transfer out rate of 29.23% (3,709.62 km2), of which the highest percentage (47.97%) is converted to grassland. There are 872.61 km2 converted to construction land. The next most degraded area is grassland, with the most significant area transferred out (10,848.23 km2), 66.97% transferred to forest land (led by natural succession), and 20.58% transferred to cultivated land (driven by agricultural expansion). Regarding transfer-in rates, construction land continues to expand, with a transfer-in rate of 227.14% (2,409.47 km2). Grassland had the most transferred area (8,539.09 km2), of which 77.40% came from forest areas. In summary, the land conversion pattern in the basin over the past 30 years was characterized by two main features: firstly, the construction land area increased exponentially; secondly, there was a frequent interchange between forest land and grassland, with forest land converted to grassland of 6,609.17 km2, while grassland was converted to forest land of 7,264.69 km2, but in general, the loss of grassland area was more significant. Over the past 30 years, the Jinsha River basin in Yunnan Province has faced the double challenge of rapid urban expansion and significant grassland degradation.

Table 4 Matrix table of land use area transfer in study area from 1990 to 2020 (km2).

1990	2020	
	Cultivated land	Forest	Grassland	Water body	Construction land	Unused land	Transferred-out area	
Cultivated land	8,982.68	974.50	1,779.52	82.17	872.61	0.82	3,709.62	
Forest	1,560.32	96,621.97	6,609.17	143.59	386.79	3.42	8,703.30	
Grassland	2,232.92	7,264.69	41,368.42	202.48	1,131.53	16.61	10,848.23	
Water body	35.16	37.36	75.20	1,363.66	15.21	1.88	164.82	
Construction land	22.15	4.15	59.89	1.00	973.24	0.37	87.56	
Unused land	0.03	0.75	15.31	5.61	3.33	141.60	25.03	
Transferred-in area	3,850.58	8,281.45	8,539.09	434.85	2,409.47	23.10		

Figure 6 Land use area transfer in the Jinsha River Basin in Yunnan Province (Note: area transferred out on the left, area transferred in on the right km2).

Characteristics of spatial and temporal carbon storage changes in the study area, 1990–2020

Changes in land use types affect carbon storage in terrestrial ecosystems in the watershed. The distribution raster map of carbon storage in the Jinsha River Basin of Yunnan Province from 1990 to 2020 was obtained by using the carbon module of the InVEST model to run and process (Fig. 7). The carbon storage values of each land type in each year were calculated (Table 5). Consistent with the pattern of the proportion of area values, the sum of the carbon storage of three land types, namely, cultivated land, forest land and grassland, accounted for the majority of the total carbon storage, with a proportion of more than 99.50% in all years. In contrast, the sum of the carbon storage of water, construction land and unused land accounted for only 0.50%. The carbon storage in the Jinsha River Basin of Yunnan Province gradually decreased during 1990–2020, with an overall decrease of 47.52 × 106 t. In terms of state and municipal scales, the distribution of carbon storage in the Jinsha River Basin of Yunnan Province shows a specific pattern, with the carbon storage contents of each state and municipal city, in descending order, as follows: Chuxiong Yi Autonomous Prefecture (CX), Dali Bai Autonomous Prefecture (DL), Qujing Municipality (QJ), Diqing Tibetan Autonomous Prefecture (DQ), Zhaotong Municipality (ZT), Lijiang Municipality (LJ), and Kunming Municipality (KM). Among them, Chuxiong Yi Autonomous Prefecture has the highest carbon storage. As the economic leader of the basin, Kunming has one of the highest levels of urban construction and, therefore, became the city with the most serious loss of carbon storage, with a reduction of 11.76 ×106 t.

Figure 7 Distribution of carbon storage in the Jinsha River Basin of Yunnan Province from 1990 to 2020.

Table 5 Carbon storage in the Jinsha River Basin of Yunnan Province from 1990 to 2020/106 t.

Land use type	1990	2000	2010	2020	
Cultivated land	296.07	290.16	293.52	299.34	
Forest	2,778.45	2,802.52	2,764.04	2,766.83	
Grassland	1,173.28	1,131.94	1,143.55	1,121.22	
Water body	0.56	0.59	0.60	0.66	
Construction land	5.84	11.81	16.22	18.63	
Unused land	0.04	0.04	0.04	0.04	
Sum	4,254.24	4,237.06	4,217.96	4,206.72	
CX	712.18	710.53	707.96	706.41	
DL	701.08	698.31	694.71	691.86	
QJ	694.29	689.78	686.36	687.15	
DQ	582.43	581.92	581.33	579.97	
ZT	555.95	555.03	553.81	551.26	
LJ	512.04	510.74	508.57	505.57	
KM	496.04	490.50	484.97	484.27	

An in-depth analysis of the raster map of carbon storage dynamics in the Jinsha River Basin of Yunnan Province in 1990 and 2020 (e.g., Fig. 8) shows that the trend of carbon storage changes in the Jinsha River Basin over a three-decade-long period is mainly characterized by stabilization with a slight decline, while growth is sporadic. This finding emphasizes the coexistence of long-term stability and local vulnerability of carbon storage in the region. Further comparing the distribution of carbon storage over time, the present study finds that the areas of carbon storage reduction are highly coincident with the degradation of forest and grassland. This correlation strongly suggests that the decrease in forest and grassland areas has a direct and decisive impact on the decline of carbon storage.

Figure 8 Changes in the spatial distribution pattern of carbon storage in the Jinsha River Basin of Yunnan Province from 1990 to 2020.

Multi-scenario land use projections for the study area, 2030

By predicting the land use distribution pattern of the Jinsha River Basin in Yunnan Province in 2030 under different scenarios, analyzed in conjunction with Table 6 and Fig. 9, under the ND scenario, as shown in Fig. 9A, the area of cultivated land continues to increase, with a predicted growth of 191.97 km2 by 2030 due to the lack of implementation of any conservation measures; the area of forest land rebounds slightly, with an increase of 38.54 km2; the area of grassland significantly decreased by 795.97 km2; water area continued to increase, increasing by 168.84 km2 under inertial development; construction land area followed the historical development pattern, increasing by 376.22 km2; and unused land area increased by only 20.41 km2.

Table 6 Land use area under different projection scenarios in Jinsha River Basin, Yunnan Province, 2030.

Land use type	ND scenario	EP scenario	FP scenario	ED scenario	
	Area/km2	Proportion %	Area/km2	Proportion %	Area/km2	Proportion %	Area/km2	Proportion %	
Cultivated land	13,025.22	7.53	12,802.19	7.40	13,748.63	7.95	12,979.34	7.50	
Forest	104,941.97	60.66	105,112.82	60.76	104,983.62	60.69	104,914.46	60.65	
Grassland	491,11.53	28.39	49,580.65	28.66	48,566.13	28.07	48,968.28	28.31	
Water body	1,967.35	1.14	1,969.76	1.14	1,937.77	1.12	1,965.44	1.14	
Construction land	3,758.93	2.17	3,388.21	1.96	3,568.78	2.06	3,978.13	2.30	
Unused land	185.11	0.11	136.47	0.08	185.18	0.11	184.47	0.11	

Figure 9 Forecast of land use distribution by scenarios in Jinsha River Basin, Yunnan Province, 2030.

Under the EP scenario, as shown in Fig. 9B, the land use strategy is significantly adjusted to enhance ecological conservation. The possibility of shifting construction land and unused land to forest land is raised, increasing construction land area much smaller than the increase in the ND scenario, with a slight increase of only 5.50 km2. In contrast, the change in unused land area is the opposite of the ND scenario, with a decrease of 28.23 km2 compared to 2020. This scenario strictly protects ecological land such as forest land, grassland and water body. It prohibits their conversion to cultivated land, which makes the change in cultivated land area also contrary to the ND scenario, decreasing by 31.06 km2 instead of increasing, making it the only scenario in which cultivated land area decreases.

Under the FP scenario, as shown in Fig. 9C, the area of cultivated land appears to be at a new high, with an increase of 915.37 km2 compared to 2020, and also an increase of 723.41 km2 compared to the ND scenario. It shows that reasonable measures and policy implementation have a significant effect on protecting cultivated land and reducing the diversion of cultivated land. Strictly controlling the conversion of cultivated land into “non-agricultural” or “non-food” land has effectively curbed the loss of cultivated land.

Under the ED scenario, as shown in Fig. 9D, the area of construction land increases significantly, by 595.42 km compared with 2020, and also by 219.20 km2 compared with the ND scenario. This change exacerbates the conversion of cultivated land, forest land and grassland to construction land. Specifically, the increase in the area of cultivated land and forest land decreases compared to the ND scenario, with an increase of only 146.08 km2 and 11.04 km2, respectively, compared to 2020, while the decrease in the area of grassland is even greater, with a decrease of 939.23 km2. In addition, the increase in the area of water body and unused land is also less than in the ND scenario due to allowing the transfer of these two land categories to other land categories.

Assessment of projected multi-scenario carbon storage in the study area, 2030

The land use pattern data simulated by the PLUS model under four different scenarios in the Jinsha River Basin of Yunnan Province for 2030 were processed, and their carbon storage was calculated using the InVEST model (shown in Fig. 10). It reveals the numerical changes of carbon storage under different scenarios and intuitively displays the spatial distribution characteristics of carbon storage.

Figure 10 Projected spatial distribution of carbon storage in the Jinsha River Basin of Yunnan Province by scenario, 2030.

Carbon storage in the Jinsha River Basin of Yunnan Province shows a continuous decreasing trend from 1990 to 2020, and it is predicted that the decreasing trend will continue until 2030 under four different development scenarios. However, the decreasing rate will be different depending on the scenarios shown in Table 7. The most drastic decrease in carbon storage will occur under the ED scenario. In contrast, the decline in the other three scenarios is relatively moderate. As shown in Fig. 11, the projected rate of carbon storage reduction under the ND scenario from 2020 to 2030 maintains a high degree of consistency with the previous actual rate of reduction from 2010 to 2020, suggesting a continuation of the rate of carbon storage reduction under the no-rule constraints. The EP and FP scenarios, with their greater focus on sustainable development and the conservation of natural resources, have seen their carbon storage reductions slowed significantly. However, under the dominance of the ED model, the blind expansion of construction land and unused land due to insufficient consideration of ecological environment and resource protection has led to the further aggravation of the rate of carbon storage loss. In the future development plan, it is necessary to prudently seek a delicate balance between economic development and ecological protection to realize the effective management of carbon storage and sustainable development.

Table 7 Carbon storage in land use patterns under different projection scenarios for the Jinsha River Basin in Yunnan Province, 2030/106 t.

Land use type	ND scenario	EP scenario	FP scenario	ED scenario	
Cultivated land	303.81	298.61	320.69	302.74	
Forest	2,767.84	2,772.35	2,768.94	2,767.12	
Grassland	1,103.34	1,113.88	1,091.09	1,100.12	
Water body	0.72	0.72	0.71	0.72	
Construction land	20.70	18.66	19.65	21.91	
Unused land	0.05	0.03	0.05	0.05	
Sum	4,196.47	4,204.26	4,201.13	4,192.66	

Figure 11 Evolutionary trends of carbon storage in the Jinsha River Basin, Yunnan Province, 1990–2030.

Figure 12 shows the pattern of carbon storage changes in the Jinsha River Basin of Yunnan Province for the four projection scenarios from 2020–2030. Under the ND scenario (Fig. 12A), the carbon storage in the study area continues to decrease. This decreasing trend is mainly concentrated in the northern region of Diqing Tibetan Autonomous Prefecture, which is mostly covered by grasslands and forests. This is in line with the gradual decrease of ecological land under the study area’s inertial development. Under the EP scenario (Fig. 12B), carbon storage remains largely stable under this scenario, with minimal areas of decrease, which matches the data that overall carbon storage shows only a slight decline, and this scenario demonstrates the positive effect of ecological protection measures in maintaining carbon storage. The FP scenario (Fig. 12C) shows a similar trend of carbon storage reduction as the ND scenario, but the magnitude of the reduction is relatively small. The ED scenario (Fig. 12D) shows clear signs of a significant reduction in carbon storage in the central regions with accelerated urbanization, especially in the Kunming and Qujing regions, highlighting the potential negative impacts of economic activities on carbon storage.

Figure 12 Changes in carbon storage distribution patterns under different scenarios in the Jinsha River Basin, Yunnan Province, 2020–2030.

Analysis of drivers of land use change and carbon storage in the study area

The LEAS module of the PLUS model can determine the degree of contribution of each factor based on the provided data on the land use distribution pattern in the two different years of the period and the data on the driving factors. Based on this, we analyzed the degree of influence of each factor on the evolution of land types during the 30 years 1990–2020 (Fig. 13). DEM is the most important driver for cultivated land, with a contribution of 0.13. This is closely related to the high topography of the study area watershed, the significant vertical climatic differences, and the crop-growing conditions. For forests, water bodies, and unused land, the most significant driving factor also was DEM, reflecting the land-use limitations imposed by the topography and geomorphology of the mountainous plateau in Yunnan Province, where the study area is located. For construction land, population, distance from highways and distance from residential areas are the main driving factors, reflecting the dependence of urban development on human resources and transportation conditions. For grassland, the population factor becomes the dominant factor, and the decrease in grassland area is closely related to human activities, especially the trend of grassland being transformed into construction land, which is significant. Natural and socio-economic factors influence land use patterns in the Jinsha River Basin of Yunnan Province, with DEM and population being the most central driving factors.

Figure 13 Drivers affecting the expansion of each taxon in the Jinsha River Basin, Yunnan Province.

The intrinsic driving mechanism of these factors on the carbon storage changes in the study area was further explored through an in-depth analysis of their contribution to the development of different land classes of area. The OPGD model’s experimental results showed significant differences in the adaptability of different driving factors to the discretization methods. As shown in Fig. 14, these methods obtain the optimal combination of the discretization parameters of each driver.

Figure 14 Combination of discretized parameters and q-value changes for each driver factor.

(1) Continuous variable discretization

The OPGD model can achieve the automated optimal discretization of the 12 driver factors continuous variables under the condition of excluding human intervention and accurately matches the optimal discretization method and the corresponding number of breakpoints for each variable. As shown in the results for Table 8, four variables, X1, X4, X5, and X8, apply to the standard deviation breaks interruption method, and six variables, X2, X3, X6, X9, X10, and X11, apply to the quantile breaks interval interruption method. The remaining X7 and X12 are suitable for the natural breaks classification method. Also, by setting the breakpoint range to 4–8 categories, the detector will select the optimal breakpoints for each factor, resulting in a higher q-value. The results show that the optimal number of breakpoints for these 12 factors matches 6–8. Through the continuous variable discretization of the model, both the accuracy and rationality of the classification are ensured, and the best classification effect is achieved.

Table 8 Parameter discrete classification.

Factor number	Factor name	Interruption method	Interrupt count	
X1	Distance from highway	standard deviation	7	
X2	Precipitation	quantile breaks	6	
X3	Population	quantile breaks	7	
X4	Distance from railway	standard deviation	8	
X5	Distance from the water system	standard deviation	6	
X6	Slope	quantile breaks	8	
X7	Temperature	natural breaks	7	
X8	Distance from the residential area	standard deviation	6	
X9	GDP	quantile breaks	7	
X10	Distance from the secondary road	quantile breaks	8	
X11	DEM	quantile breaks	6	
X12	Soil type	natural breaks	8	

(2) Single factor detection

Based on the Optimized Parameter Geodetector (OPGD) model, the explanatory power (characterized by q-value) of 12 key drivers of carbon storage dynamics in the Jinsha River Basin of Yunnan Province was systematically quantified. The analysis results, as shown in Fig. 15, show that the explanatory power of each driver exhibits significant differences. Regarding the dominant natural drivers, the temperature factor has the most prominent explanatory power (q = 0.70), indicating that the temperature condition is the primary natural factor regulating the spatial differentiation of carbon storage in the basin. The explanatory power of the topography factor (DEM, q = 0.45) and precipitation factor (q = 0.40) exceeded 40%, confirming the coupled topography-climate effect’s key role in forming carbon storage patterns. As for the important socio-economic drivers, the explanatory power of GDP (q = 0.38), population (q = 0.35) and distance to highway (q = 0.33) exceeded 30%, reflecting the significant influence of human activities and land-use patterns on carbon storage. In this study, the analysis of driving mechanisms was conducted by optimizing the breakpoint settings several times and conducting sensitivity analyses. It was finally determined that air temperature, DEM, and precipitation were the core drivers of carbon storage changes in the basin. These three are all-natural factors, indicating that in the ecologically fragile Jinsha River basin, the fundamental regulatory role of natural geographic elements is stronger than socioeconomic factors.

Figure 15 Interpretation of carbon storage changes in the study area by single and interactive factors.

(3) Interaction factor detection

Interaction detection analysis revealed the compound driving mechanism of carbon storage changes in the Jinsha River Basin of Yunnan Province, as shown in Fig. 15. The results show that the interactions among the driving factors significantly enhance their ability to explain the carbon storage changes. The interaction effect of the temperature factor is particularly prominent, showing typical coupling characteristics. The explanatory power of the interaction between temperature and population factor reached 0.84, indicating that the effect of human activity intensity on carbon storage was significantly amplified in the context of climate warming. The q-value of the temperature and slope factor interaction reached over 0.82, confirming that topographic conditions significantly affected vegetation productivity and soil organic carbon decomposition rate by regulating the heat redistribution process. The q-value of the interaction between air temperature and the distance from the residential area factor was also above 0.82, reflecting that the spatial pattern of the intensity of human activities decreasing with distance plays an important role in regulating the distribution of carbon storage. All interactions showed two-factor or nonlinear enhancement effects, and no independent or attenuated effect types were found. The results of this study suggest that (1) there are complex non-additive effects among the driving factors, (2) changes in carbon storage in the watershed are the result of the synergistic effects of multiple environmental factors, and (3) traditional one-way analyses may underestimate the actual extent of the effects.

Integrated analysis employing the LEAS module (PLUS model) and OPGD model reveals distinct mechanistic pathways governing land use expansion and carbon storage evolution. The LEAS module analysis (Fig. 13) identifies X11 (DEM) and X3 (population) as dominant contributors during the expansion of six major land types, demonstrating their pivotal role in regional land use transformation. The OPGD model quantification (Fig. 15) establishes X7 (temperature) with maximal independent explanatory power for carbon storage (q = 0.70), followed by X11 (q = 0.45), while interaction analysis reveals optimal synergistic effects between X3 (population) and X7 (temperature) (q = 0.84).

These mechanistic divergences originate from fundamental analytical scale differences: LEAS emphasizes localized land-type responses whereas OPGD targets system-level carbon dynamics. Despite incomplete correspondence in explanatory significance between land use expansion drivers and carbon storage influencing factors, a pronounced spatial coupling relationship is established. For instance, the DEM factor demonstrates high explanatory power in both modules. This dual prominence directly correlates with the Jinsha River Basin in Yunnan Province’s remarkable topographic heterogeneity (>3,000 m elevation gradient), which simultaneously regulates both land use suitability and carbon sequestration potential.

Discussion

Analysis of drivers of land class expansion and ecosystem carbon storage in the Jinsha River Basin, Yunnan Province, China

Analysis of the contribution of driving factors based on the PLUS model showed that land use changes in the Jinsha River Basin of Yunnan Province were mainly significantly influenced by topographic (DEM) and demographic factors. Among them, the DEM factor dominates, mainly because the study area is dominated by a mountainous plateau (>50% of the land with slope >15°), and topographic constraints become a key factor limiting land development and expansion. In addition, the impact of population density on cultivated land expansion was particularly prominent, consistent with Chen & Yao’s, (2023) findings in Mashan County, which indicated that the population growth driver was the core driver of cultivated land expansion. Among the 12 driving factors, the influence of GDP is generally high, indicating that regional economic development profoundly impacts the evolution of land use patterns. In economically developed regions, land use changes are more frequent, often accompanied by the expansion of construction land and the over-consumption of ecosystem services; in economically underdeveloped regions, land development is restricted, and GDP growth has become a key factor driving land use transformation (Luo & He, 2023).

The single factor detection analysis of the optimal geoprobe showed that temperature (q = 0.70), DEM (q = 0.45) and precipitation (q = 0.40) were the strongest natural factors influencing the spatial differentiation of carbon storage, which was highly consistent with the findings of the global mountain ecosystems study by Sun et al. (2022), and further verified the strong association between elevation gradient, hydrothermal conditions and carbon density were strongly correlated. The interaction detection analysis reveals that the interaction between temperature and population density (q = 0.84) has the strongest explanatory power for carbon storage changes. This indicates that climate-human activity synergy is the central mechanism driving carbon storage evolution, consistent with the study of Li et al. (2023a). This is manifested in the increase in food demand due to population growth and conversion of forest/grassland to cultivated land, with a total of 1,560.32 km2 of forest land and 2,232.92 km2 of grassland converted to cultivated land in the study area from 1990 to 2020, leading to a decrease in soil organic carbon density; On the other hand, population growth makes ecological land more patchy through the spatial heterogeneity diffusion of urbanization. At the same time, the increasing construction land will also exacerbate the fragmentation of the landscape pattern and indirectly reduce the function of carbon sinks (Feng et al., 2020).

Impact of land use change on carbon storage in the Jinsha River Basin, Yunnan Province, China

At the global scale, land use change contributes about 14% of total carbon emissions, and this proportion may be higher in ecologically vulnerable areas. In this study, we focus on the land use transition process in the Jinsha River Basin of Yunnan Province from 1990 to 2020 and find that the carbon storage in the region shows a significant downward trend (cumulative decrease of 47.52 × 106 t), and the main driving mechanisms are the loss of high carbon density land (forest and grassland) due to the expansion of land for construction, the fragmentation of the landscape triggered by the acceleration of the urbanization process, and the attenuation of soil organic carbon brought about by the intensification of agriculture. This finding echoes spatially with Liang, Liu & Huang’s (2017) study of oases in Northwest China and Arunyawat & Shrestha’s (2016) study assessing the ecological impacts of land use change in Northern Thailand, which together corroborate the carbon sink loss effect of ecological land conversion in the context of rapid urbanization. The four types of development scenarios for 2030 constructed based on the PLUS-InVEST coupled model show that the EP scenario retains 7.79 × 106 t more carbon storage compared to the ND scenario through the implementation of a strict ecological red line policy. The analysis of spatial heterogeneity shows that Chuxiong Yi Autonomous Prefecture realizes carbon storage gain under the FP scenario. In contrast, Dali Bai Autonomous Prefecture and Qujing Municipality realize carbon sink increases under both EP and FP scenarios. This positive result is attributed to the fact that these three regions are relatively rich in cultivated land area and possess high-quality ecological environments, which enable them to better maintain and enhance carbon storage in the face of rule shifts in different land classes. Diqing Tibetan Autonomous Prefecture, Zhaotong City, and Lijiang City show significant carbon storage declines in all simulation scenarios, highlighting the sensitivity of high-altitude ecoregions to the expansion of construction land. In addition, Kunming City in the Economic Circle has successfully realized the carbon storage recovery under the EP scenario by slowing down the construction and development of the urban area through appropriate rule adjustments. Based on the above analyses and findings, this study suggests the implementation of differentiated land management strategies in the Jinsha River Basin of Yunnan Province, the establishment of a dynamic spatial control system with “three zones and three lines”, the implementation of a cross-regional synergistic market mechanism based on the compensation of carbon sinks, and the adoption of integrated air-sky-earth monitoring, etc., which provide a generalizable technological pathway to realize the spatial management of the national territory under the goal of “dual-carbon”.

Limitations and prospects

This study found the following key limitations of the PLUS-InVEST-OPGD model in the application experiments in the Jinsha River Basin of Yunnan Province (covering seven states and municipalities). The variability of development planning in each state and municipality makes it difficult to accurately reflect the standard conversion rules (e.g., cost matrix settings), and regional adaptability is limited. It is challenging to capture small-scale land use changes in the study area using spatial resolution (30 m) land use data. The partial absence of carbon pools in water body and unused land and the uncertainty in the spatial and temporal variability of carbon density are prone to carbon storage assessment bias (Li et al., 2020). Future research can adopt the coupling of deep learning and mechanistic modeling, develop a new type of Transformer-MCCA model, and adopt multi-source data fusion technology to reduce the assessment error of the model and further enhance its simulation accuracy and refinement. Based on the 14th Five-Year Plan (2021–2025) of Yunnan Province, we embedded key policy variables, constructed a policy response module, and developed a policy-driven model optimization framework to provide a reference scheme for the sustainable development of other watersheds of the same type.

Conclusion

In this study, we systematically analyzed the mechanism of land category evolution on carbon storage based on the land use/cover change (LUCC) data of the Jinsha River Basin in Yunnan Province from 1990 to 2020. We constructed four development scenarios (ND, EP, FP, and ED) based on sustainable development to predict the carbon storage change in 2030. The main research findings are as follows: (1) Between 1990 and 2020, the land-use pattern of the Jinsha River Basin in Yunnan Province underwent a remarkable transformation, characterized by the spatial reconfiguration of the “ecology-production” dichotomy. Among them, the grassland area decreased by 2,317.10 km2 (annual average decrease of 77.24 km2), which was mainly transformed into forest and cultivated land. The construction land expansion is 2,321.91 km2 (average annual growth of 77.40 km2), which is mainly converted from grassland and cultivated land. This trade-off between “grassland shrinkage and construction land expansion” reflects the land use conflicts in the region’s rapid urbanization process. (2) Between 1990 and 2020, the carbon storage in the watershed showed a continuous downward trend, with the total amount falling from 4,254.24 ×106 t (1990) to 4,206.72 ×106 t (2020), with a net loss of 47.52 ×106 t. The structure shows the transformation of “high-carbon land (forest land and grassland) to low-carbon land (construction land and cultivated land)” and this land transformation contributes 32% of the total carbon loss. (3) Scenario simulation in 2030 shows that the carbon storage in the ND scenario drops to 4,196.47×106 t; the EP scenario protects the natural ecological environment through practical constraints on economic activities, resulting in a relatively small reduction in carbon storage, and eventually, there is carbon storage of 4,204.26×106 t; and the FP scenario reduces the carbon storage in a relatively moderate way, with carbon storage of 4,201.13×106 t; In contrast, the ED scenario has a large-scale expansion of construction land and a large amount of ecological land is occupied, resulting in a decrease in carbon storage to 4,192.66×106 t. (4) The DEM and population factors significantly contribute to the evolution of the six land categories, which are the core drivers affecting the LUCC. The OPGD model reveals that temperature as a key climate factor (q = 0.70) and its interaction with the population factor (q = 0.84) form a coupled “climate-human” driving mechanism, and this coupling greatly improves the explanatory power, indicating that the impact of human activities on the carbon cycle is significantly amplified in the context of climate change.

Supplemental Information

Supplemental Information 1 Land use data in the research area in 1990

Raster data obtained by cropping and reclassifying the original data downloaded from the official website into first level land classes. This is also the basic data for our experiment, and almost all experimental processes were conducted based on this data.

Supplemental Information 2 Land use data in the research area in 2000

Raster data obtained by cropping and reclassifying the original data downloaded from the official website into first level land classes. This is also the basic data for our experiment, and almost all experimental processes were conducted based on this data.

Supplemental Information 3 Land use data in the research area in 2010

Raster data obtained by cropping and reclassifying the original data downloaded from the official website into first level land classes. This is also the basic data for our experiment, and almost all experimental processes were conducted based on this data.

Supplemental Information 4 Land use data in the research area in 2020

Raster data obtained by cropping and reclassifying the original data downloaded from the official website into first level land classes. This is also the basic data for our experiment, and almost all experimental processes were conducted based on this data.

The authors thank those who provided methodological and writing guidance that made this study possible.

Additional Information and Declarations

Competing Interests

Author Contributions

Data Availability

The authors declare there are no competing interests.

Lichang Huang conceived and designed the experiments, performed the experiments, analyzed the data, prepared figures and/or tables, and approved the final draft.

Xue Ding conceived and designed the experiments, authored or reviewed drafts of the article, and approved the final draft.

Jinliang Wang conceived and designed the experiments, authored or reviewed drafts of the article, and approved the final draft.

Shuangyun Peng conceived and designed the experiments, authored or reviewed drafts of the article, and approved the final draft.

The following information was supplied regarding data availability:

The original data is available at the Aerospace Information Research Institute,Chinese Academy of Sciences: http://www.aircas.ac.cn.

The first set of global 30 meter resolution land cover dynamic products (GLC_FCS30D) is available at the International Research Center of Big Data for Sustainable Development Goals-CBAS and Zenodo:

- https://data.casearth.cn/thematic/glc_fcs30/314.

- Liangyun Liu, Xiao Zhang, & Zhehua Li. (2025). GLC_FCS30D: the first global 30-m land-cover dynamic monitoring product with fine classification system from 1985 to 2022 (Version v2) [Data set]. Zenodo. https://doi.org/10.5281/zenodo.15063683.

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
