# Peer review of "Using PLUS-InVEST-OPGD model to explore spatiotemporal variation of ecosystem carbon storage and its drivers in Jinsha river basin, China"

_PeerJ, doi:10.7717/peerj.19681_

## Round 0.1 · original submission · Major Revisions

Dear authors, I ask you to carefully correct the manuscript and hope that the new version of this article will be approved by the reviewers.

Reviewer 1 ·

Basic reporting

The paper optimized regional carbon storage by integrating the optimal parameter geographic detector with the PLUS-InVEST model. The authors clearly articulate the advantages and operational mechanisms of three models in comparison to previous models, and conduct a spatiotemporal dynamic analysis of carbon storage using raster data. The study demonstrated the effectiveness of this approach in ecological conservation and carbon storage management, providing valuable insights within this field.

Experimental design

The research methodology of the manuscript is clearly articulated, and the dataset employed for method validation is appropriate. The paper utilizes standard evaluation metrics, ensuring the comparability of the results.

Validity of the findings

The results of the paper have a certain degree of reliability and are supported by sufficient data.

Additional comments

1) The abstract fails to highlight the key statement regarding the minimal reduction of carbon storage in the ecological protection context. It is suggested that the abstract explicitly indicate that the core of the study is the effective control of carbon storage in this context.
2) The paper does not provide a detailed explanation of the rationale behind the selection of driving factors. Please clearly explain the basis for selecting these factors in the text to enhance the rigor of the analysis.
3)It is suggested to add a discussion on how the PLUS model meets accuracy requirements, and include supporting relevant literature.
4)When describing the prediction results of various scenarios, it is recommended to compare them not only with the 2020 outcomes but also with the 2030 natural development scenario. This approach will help to highlight the strengths and weaknesses of each scenario, offering a more comprehensive analysis.
5)The discussion section requires further elaboration. It is suggested to explore the effects of specific scenarios, rather than merely generalized trends and policies. Each proposed scenario’s fundamental principles should be examined in detail, analyzing their impact on policy effectiveness, comparing various scenarios, and proposing developmental recommendations.
6)There is an inconsistency in the figure legend numbering. Please verify the figure legend numbers throughout the text and correct them.
7)It is recommended to standardize the color scheme for land use types in the pie chart and the legend in Figure 4, and to lighten some of the darker colors in the legend to prevent data from becoming indistinct. Similarly, Figure 9 exhibits similar issues and should be modified accordingly.
8)Please verify the units in the legends of Figures 7 and 12 to ensure they should read carbon density rather than carbon storage, and confirm the accuracy of the units.

Reviewer 2 ·

Basic reporting

This paper meets the professional standards of courtesy and expression, and include sufficient introduction and background and relevant prior literature.
This paper simulates land use change under four different scenarios in the future based on PLUS-InVEST-OPGD, estimates carbon stocks, and reveals the driving factors. The content of the article is relatively rich, and the method and structure are basically complete. However, there are still many problems that need to be solved.

Major comments:
1.The logic of research methodology content is confusing. The methodology used by the authors is mature and well-established, the research method should be short, and your manuscript contains a lot of information about the model, which is more like an introduction.
2. The names of land types are not uniform. For example, forest land and cultivated land, according to the corresponding classification in the article, should be forest land, cultivated land, but the text appears woodland, arable land, please carefully check and check the name of other land use types.
3. The language of the article should be concise and professional, and the reference citation should not be standardized. There are many paragraphs in the article that are not concise enough, and the descriptions are too long and complex. Secondly, there are more misspellings of words. The citation format of references should be revised according to the requirements of the journal.

Minor comments

Abstract
Please reorganize the Abstract. The language is too verbose and complex, so simplify the background statement. In addition, it should be noted that the abstract is a refinement and summary of research results, expressing the author's key conclusions, rather than simply listing them.

Introduction
1.It is suggested that the author should add a discussion on the importance of carbon stock simulation and prediction in paragraph 3 of the introduction.
2. Please add references to support the author's discussion of the current status of the research site in the fifth paragraph of the introduction.

Experimental design

The research question is well defined, relevant & meaningful, and within aims and scope of the journal.The methods are described with sufficient information. However, there are still some shortcomings in the detailed description of some research and analysis methods:

Data set
1. It is suggested that the author improve Table 1 to specify the source, format, precision, and purpose of the data, especially to indicate that the data is used for a specific part.
2. Please check and confirm whether the legend in Figure 2 is correct ( temperature -67.86-234.56°C ? )

Research framework
1.Here the author draws a clear and beautiful research framework (Figure 3) to express the research idea, while the text description looks more like a methodology. I suggest the author briefly summarize the core points.

Research Methodology
1.The author presents the relevant results of land use transfer in the Results, and proposes to add relevant analysis steps of land use transfer to the methodology (For example, the land use transition matrix).
2. Here, the author has designed four scenarios: ND, EP, FP, and ED. Among them, EP, FP and ED are set by the authors in the ND scenario, and the authors are requested to specify the basis for setting the relevant parameters in each scenario, such as adding references to refer to the relevant research of others in the region, or the relevant national policy basis.
3. It is recommended that the author reorganize the structure of the methodology section, "Setting of different future development scenarios" should be part of the "PLUS model". The PLUS simulation analysis process should be "LEAS" - "CARS" - "Model accuracy verification" (2020) - "Set different future scenarios".
4.When analyzing the driving factors, the results of factor detection at different scales will vary greatly. It is necessary for the author to specify the sampling intensity in the process of factor detection, whether it is sampled according to "city-state", "county-district" or "kilometer grid"?

Validity of the findings

The study used conventional methods to conduct research and analysis, and obtained corresponding results. No special innovation was involved. However, the necessary processes that are missing need to be carefully supplemented to ensure that the analysis results are more realistic.

Results
1.Line480-488, this passage describes the carbon stocks of different states, but there should be corresponding data to support it;
2.Line506-508, please reorganize this sentence to indicate the order;
3.Line517-526, Here, the authors use the actual land use data in 2020 to compare with the simulation data to verify the accuracy of the model prediction, which is very important in the simulation and forecasting process. But we generally think that this should fall under the methodological "PLUS" simulation prediction part, not the result;
4.Line537-538, the description here should highlight the characteristics of your study area and not Yunnan Province;
5.Line594, Figure 12, the spatial distribution of carbon stocks under the four scenarios does not seem to be different, it is recommended to zoom in to highlight the key different regions, such as Figure 9;
6.Line630-632, Please specify how the 12 drivers were screened and identified.
7.Line635-640 and 644-657, this paragraphs are more like Methodology, which are the research process rather than the result.
8.Completeness of supplementary result data, , not seen in your manuscript.
9.Line661, line678 (Fig17) do not appear in the text, please add the completeness of the resulting data.

Additional comments

Discussion
1.The authors fully discussed the driving factors of land use transfer and the influencing factors of carbon storage respectively in the discussion, which is worthy of recognition. However, what we would like to see more is that the authors used different methods (LEAS and OPGD) to obtain the linkages and differences between the influencing factors of land use transfer and carbon stocks. The author also mentions that the core reason for the change of ecosystem carbon storage is the transfer of surface land use types, and there should be a direct relationship between the factors affecting land use transfer and the factors affecting the change of carbon stock.
2. Line768-777, It is recommended that the author should elaborate on the response of this study to national policy in context.

Reviewer 3 ·

Basic reporting

The literature review in the article is generally up-to-date and well-structured, as it provides an overview of key studies on the impact of land use change on carbon storage in ecosystems. The authors of the article refer to a significant number of scientific sources, covering both classical works on the assessment of changes in carbon stocks and modern studies related to the use of models, such as InVEST, PLUS and OPGD. This demonstrates the authors' attempts to integrate the latest methodological approaches to analyse spatial and temporal changes in carbon stocks.
However, despite the high level of detail, there are several aspects that may affect the relevance of the literature review. First, although the authors mention important studies on anthropogenic carbon stock changes, some of the sources used are relatively outdated or do not take into account the latest trends in the use of modelling to predict changes in landscapes. For example, most of the studies mentioned relate to the period up to 2022, although recent years have seen a significant development of approaches to analysing ecosystem services using machine learning and big data, which should be taken into account.
Second, the article is dominated by a quantitative analysis of the literature without sufficient critical reflection on the limitations of the approaches used. Although the authors compare different methods of predicting land use change and its impact on carbon stocks, they do not focus on the potential biases of the models, which could have made their analysis more balanced. For example, a discussion of the limitations of the InVEST model, such as its inability to account for short-term changes in the carbon balance, could have added depth to the analysis.
In addition, while the review mentions studies on carbon stock changes in river basins in other regions, it does not always make a clear connection to the specifics of the Jinsha River Basin. It would be useful to make a more detailed comparison with similar regions with similar natural and socio-economic conditions to emphasise the uniqueness of the authors' approach.
Thus, the literature review of the article is generally relevant, as it takes into account the main theoretical foundations and modern approaches to the study of changes in carbon stocks. At the same time, it could be supplemented by a deeper critical analysis of the models, the use of the latest research, and an expanded comparison with other regions to more fully reveal the scientific context of the study.

Experimental design

The experimental design of the article has several drawbacks that may affect the accuracy and generalisability of the findings. The main limitation is the use of the PLUS model for forecasting land use changes, which is based on specified development scenarios. This means that the results obtained are largely dependent on assumptions about future changes in political, socio-economic and environmental conditions, which can lead to significant uncertainty in the projections. At the same time, the assessment of changes in carbon stocks is carried out using the InVEST model, which, although an effective tool for analysing ecosystem services, has certain limitations. The authors acknowledge that this model is not able to accurately estimate carbon stocks in catchment areas and on unoccupied land, which can lead to a distortion of the overall picture of changes in the region's carbon balance.
Another significant limitation is the lack of empirical field data, as all calculations are based on modelling and secondary data. The absence of direct measurements of carbon stocks in different land cover types can reduce the accuracy of forecasts and create a gap between theoretical estimates and actual conditions. In addition, the accuracy of carbon stock estimates largely depends on the initial carbon density data obtained from previous studies and adjusted according to the model. The authors note that even after adjustment, the values obtained differ from actual measurements, indicating the need for further improvement of the estimation methods.
The scenario approach used in the study allows for an assessment of possible scenarios, but it does not take into account unforeseen events such as extreme weather, policy changes or economic crises. This can affect the accuracy of forecasts, as scenarios are based on deterministic assumptions and do not take into account all possible scenarios.
Another drawback is the lack of detailed validation of the results. Despite the use of modern modelling techniques, the article does not compare the forecasts with independent observations or other models. This makes it difficult to assess the reliability of the results and does not allow us to draw conclusions about their accuracy.
Thus, although the experimental design of the article is well developed and based on modern modelling methods, it has certain limitations, including dependence on models, lack of empirical data, possible distortions due to the characteristics of the source data, and insufficient validation of the results. To improve the study design, the authors could incorporate empirical measurements of carbon stocks, use alternative models to test the predictions, and supplement the analysis with a multifactorial risk assessment to increase the reliability of the conclusions.

Validity of the findings

The validation of the conclusions in the article exhibits certain deficiencies that may compromise the reliability of the results. Firstly, while the authors provide all the raw data and these data are statistically sound, the impact and novelty of the findings have not been assessed. This means that, although the data presented is controlled and relevant to the research question, its practical relevance and potential impact on the wider academic discourse remains insufficiently addressed.
Secondly, although the conclusions are well formulated and clearly related to the research question, they are limited to the results obtained.An important shortcoming is the lack of independent verification of the results or comparison with other existing studies.For example, it would be useful to assess how the results of this work are consistent with other carbon stock assessment models or empirical studies.
Furthermore, reliance on forward-looking modelling scenarios, whilst advantageous in terms of predicting possible changes in land use and carbon stocks, remains conditional upon a number of assumptions, including the accuracy of the PLUS and InVEST models, which may have limitations in capturing complex environmental processes.
While the study's findings are statistically robust and directly pertinent to the research question, their validation could be enhanced through the incorporation of independent verification, a comparison with alternative methodologies, and a more thorough assessment of the practical implications of the findings.

Additional comments

To improve the practical value of the paper, the authors could take several key steps. First, the validation of the results should be strengthened by comparing the predictive models with empirical observations or other known methods of estimating changes in carbon stocks. The use of alternative approaches, such as remote sensing data or field measurements of carbon in soils and vegetation, would allow for a more accurate and reliable assessment of the actual state of ecosystems.
Second, the article would benefit from a broader analysis of possible development scenarios. While the authors consider several options for land use change, taking into account other factors such as extreme climate events, socio-economic changes, or ecosystem conservation policies would allow for a better assessment of the long-term implications for carbon stocks.
It would also be useful to include an assessment of the environmental and socio-economic impacts of projected changes. For example, consideration could be given to how changes in the carbon balance will affect biodiversity, the resilience of local ecosystems, and opportunities for carbon regulation at the regional level.
Another important step to increase the practical value of the paper is to develop recommendations for environmental management and policy. The authors can propose specific measures to preserve natural landscapes, optimise land use, and reduce the loss of carbon stocks in the future. This will increase the practical value of the study not only for the scientific community, but also for decision-makers in the field of environmental policy and natural resource management.
Finally, the paper could gain additional practical value through a more detailed interpretation of the risks and uncertainties associated with the projections. This will allow for a better understanding of the limitations of the models, taking into account possible deviations from the scenarios and adapting future strategies to different scenarios.
In general, supplementing the study with empirical data, expanding the scenario analysis, assessing environmental and social impacts, formulating practical recommendations, and taking into account uncertainties in the forecasts will significantly increase the practical relevance of the work and its usefulness for decision-making in the field of ecosystem management and carbon conservation.

Reviewer 4 ·

Basic reporting

The authors attempt to investigate the spatiotemporal variations in ecosystem carbon storage and its driving factors in the Jinsha River Basin, China, using the PLUS-InVEST-OPGD model. This research topic holds certain scientific and practical significance. However, there are several issues in study design, data analysis, and result interpretation that require improvement. Overall, the manuscript presents its findings in a somewhat redundant manner, and its logical structure needs enhancement. With appropriate revisions, this version could be considered for publication in PeerJ.
Major issues:
1.Issues in Model Applicability and Limitations. The study primarily employs the PLUS and InVEST models to simulate carbon storage but fails to adequately discuss their applicability and limitations. For example, the PLUS model is mainly used for land use transition simulation, yet it may have limited accuracy in capturing small-scale land use changes. Similarly, the InVEST model relies on default carbon storage parameters, which may not accurately reflect the actual carbon cycling processes in the study area. Additionally, the study does not provide a thorough assessment of model errors or uncertainty analysis, which could impact the reliability of the results. It is recommended that the authors enhance the discussion on model applicability to improve the credibility of their findings.
2.Limitations in Driving Factor Analysis. The study utilizes the OPGD model to analyze the driving factors of carbon storage but does not sufficiently explain the scientific rationale for variable selection or their explanatory power in relation to carbon storage changes.
3.Lack of In-depth Comparison with Existing Studies and Discussion of Innovation. The study references several investigations on carbon storage dynamics at the watershed scale but fails to clearly articulate its unique contributions compared to previous research. Notably, multiple studies have already applied the PLUS-InVEST model combination for carbon storage analysis. However, this study does not specify methodological improvements or novel contributions. It is recommended that the authors clarify the study’s advancements, whether in parameter optimization, higher-resolution data integration, or innovative scenario prediction approaches, to strengthen its academic contribution.

Details:
1.Line 19, big changes? Check for this colloquialism throughout the text.
2.Overall the results description section of the abstract is a bit wordy, please present the highlight results of the article.
3.The introduction part, the second paragraph expresses the land use change and the third paragraph describes the InVEST model, which is feasible overall but too cumbersome in general, please optimize the presentation, my suggestion is that the two paragraphs can be integrated.
4.In line 64, “On the contrary” applies when there is a clear contrast between the two, whereas “conversely” or “by contrast” would be more appropriate. ”
5.In the section introducing the influencing factors, the advantages of the geodetector model are only illustrated, and the lack of research on the drivers is not expressed, and I think that adding this section would highlight the innovativeness of this study more clearly.
6.Introduction section, lines 136-137, cite relevant references when describing OPGD modeling.
7.Overall the introduction section does not clearly express the necessary and innovative points of this study, please refine and add them.
8.Line 174, meters can be replaced with m, similarly synonymous.
9.In the methodology section, the description of the geodetector is too cumbersome, please simplify it.
10.In the results section, all the results are too long, please pick the important ones to present and integrate them where you can.
11.Drivers discretized why only 3 categories (4-6), more categories may have higher q-values.
12.For maps of the Jinsha River in the article, please standardize the scale, e.g., use (1,75,150 km).

Experimental design

no comment

Validity of the findings

no comment

Additional comments

no comment

Annotated reviews are not available for download in order to protect the identity of reviewers who chose to remain anonymous.

---

## Round 0.2 · Minor Revisions

Dear Dr. Huang, I ask you to correct the shortcomings pointed out by reviewer 4 and this article will be accepted for publication.

Reviewer 1 ·

Basic reporting

no comment

Experimental design

no comment

Validity of the findings

no comment

Additional comments

1.In the analysis of land use change and carbon stock driving factors in the study area (line 490), it is recommended to combine the Plus model and OPGD model for a more in-depth and comprehensive analysis of the influence of each factor on carbon stock, avoiding a simplistic presentation of the results.
2.The pie charts in Figures 5 and 9 are not sufficiently clear. It is recommended to optimize them to ensure better readability and clarity of the charts.

Reviewer 3 ·

Basic reporting

All recommendations have been taken into account. I recommend the article for publication.

Experimental design

All recommendations have been taken into account. I recommend the article for publication.

Validity of the findings

All recommendations have been taken into account. I recommend the article for publication.

Additional comments

All recommendations have been taken into account. I recommend the article for publication.

Reviewer 4 ·

Basic reporting

The authors have made commendable efforts in revising the manuscript based on the first round of review comments. Their comprehensive responses and substantial improvements have enhanced the overall quality of the paper. The revised manuscript demonstrates progress in structure, argumentation, and presentation. While the manuscript now essentially meets publication standards, I offer the following suggestions for further refinement:
1.For results related to the Geodetector analysis, the q-values should retain only two decimal places (instead of four) to align with standard reporting conventions.
2. Inconsistent formatting for figure references (e.g., "Figure 12b" vs. "Fig. 12c") should be standardized throughout the manuscript. A uniform style (either "Figure" or "Fig.") is recommended.
3.Address inconsistencies in unit notation, particularly between "tons" and "t" (e.g., lines 334 and 342). A thorough check of unit consistency across the entire manuscript is advised.
4.Ensure consistent formatting for numerical values and their associated units. For example, remove spaces between numbers and percentage symbols (e.g., "50%" rather than "50 %").
5.Line 160 contains an apparent formatting irregularity ("[27]"). Please meticulously review all in-text citations and references to ensure adherence to journal guidelines.

Experimental design

no comment

Validity of the findings

no comment

Additional comments

no comment

---

## Round 0.3 · accepted · Accept

Dear Dr. Huang, I congratulate you on the acceptance of this manuscript for publication. I hope that you will send such high-quality articles to our journal more often.

Reviewer 4 ·

Basic reporting

no comment

Experimental design

no comment

Validity of the findings

no comment

Additional comments

no comment